# A Novel Tool to Assess the Risk for African Swine Fever in Hunting Environments: The Balkan Experience

**DOI:** 10.3390/pathogens11121466

**Published:** 2022-12-03

**Authors:** Mario Orrico, Mark Hovari, Daniel Beltrán-Alcrudo

**Affiliations:** 1Food and Agriculture Organization of the United Nations (FAO), Regional Office for Europe and Central Asia, 1054 Budapest, Hungary; 2One Health Master, Utrecht University, 3584 CS Utrecht, The Netherlands

**Keywords:** African swine fever, wild boar, hunting, hunter, Balkans, risk assessment, biosecurity, expert knowledge elicitation

## Abstract

In Europe, African swine fever (ASF) can be sustained within wild boar populations, thus representing a constant source of virus and a huge challenge in the management of the disease. Hunters are the key stakeholders for the prevention, detection and control of ASF in wild boar. Their behavior and the biosecurity standards applied in infected or at-risk hunting grounds have a huge impact on disease dynamics and management. The Food and Agriculture Organization (FAO) has developed a semi-quantitative survey-based novel tool to assess the risk of ASF in hunting grounds (namely the risks of introduction and spread into and between hunting grounds, and the risk of not detecting the infection) and how such risks could be reduced if mitigation or corrective measures were applied at low, medium and high effort. The weight of risk factors was determined through an expert knowledge elicitation (EKE). The surveys for each hunting ground were filled in by their respective managers. The tool’s outputs allow users to visualize the different ASF risks of hunting grounds, whether as numerical values or color-coded maps, at sub-national, national and regional levels. These outputs can be used to guide policy makers, highlighting gaps or geographical areas to prioritize. The tool was used to assess hunting grounds in Kosovo^1^ (^1^ As per United Nations Security Council resolution 1244). Montenegro and Serbia, showing overall a high risk.

## 1. Introduction

African swine fever (ASF) is a highly contagious viral disease of domestic and wild pigs caused by a virus belonging to the *Asfaviridae* family; genus *Asfivirus*, with high lethality rates for any age and sex, which is responsible for serious economic and production losses [1]. There is currently no effective vaccine or treatment against ASF. Twenty-four genotypes are known globally, but only two are present outside Africa: genotype I, limited to the Italian island of Sardinia and Shandong and Henan in China [2] and genotype II, responsible for the recent panzootic that began in 2007 in Georgia and is currently spreading throughout Eurasia and the Caribbean.

In the Balkans, the first case of ASF was reported in July 2019 in domestic backyard pigs in the village of Rabrovac in central Serbia [3]. At the time, ASF was already present in several countries in the region, namely Bulgaria, Hungary, Romania and an isolated outbreak in domestic pigs in Greece that has since been contained. Since this first outbreak in Serbia, ASF has been (sporadically) reported in both wild boar and domestic pigs, predominantly in backyard settings, but also in a commercial pig farm in April 2021 [4]. Within the region, North Macedonia confirmed its first ASF outbreak in domestic pigs in January 2022 and, later the same year, also detected ASF in hunted and found dead wild boar and backyard pigs in the eastern part of the country bordering Bulgaria.

The presence of ASF virus in wild boar populations often leads to outbreaks in domestic pigs, usually those kept outdoors and on non-commercial farms, also known as “backyard” farms where biosecurity tends to be lowest [5], but also on commercial farms. The prolonged circulation of the ASF virus within the wild boar population has become a new transmission cycle not described before, whereby the disease can be sustained by wild boar populations without any involvement of domestic pigs [1,6]. Major risk factors for ASF spread and persistence in wild boar populations include natural wild boar movements [7], direct contact with free-roaming pigs [8], wild boar density [9], supplementary feeding [10], the presence of infected carcasses [11], improper disposal of infected pork or wild boar products, the use of non-sanitized fomites and other contaminated materials, and human-mediated transport of live wild boar [12]. These allow the virus to persist, spread and occasionally make geographic leaps, even hundreds of kilometers from infected areas. From looking at the list of risk factors, the crucial role of the human component becomes apparent.

The continuous presence of the virus in the affected wild boar habitats represents a great challenge to the pig production sector, the wildlife management authorities and the hunting sector. The experiences of infected countries when encountering ASF in wild boar have shown that immediate and coordinated implementation of control measures in wild boar habitats, while the infection is still restricted to a limited area, is crucial to have a chance at confining and eventually eradicating the disease, as happened in Belgium and the Czech Republic [13,14]. For a prompt implementation of control measures, the early detection of the virus’ entry into free wild boar populations is a crucial first step and prerequisite. A delay in detection will likely lead to a wider silent spread of the disease, hampering the control and eventual eradication of ASF.

The human–animal interface, particularly between hunters and wild boar, represents one of the most challenging but necessary aspects to consider when managing this disease. The role that hunters play in the surveillance of wild animal diseases is widely recognized by the international community [15]. Vergne et al. [16] emphasized the need to develop more effective communication strategies, targeted at hunters, about the disease, its epidemiology, consequences and control methods, to increase the likelihood of early reporting. The inclusion of hunters in designing and implementing surveillance strategies before, during and after an ASF outbreak, as well as their practices in terms of biosecurity, are determinant factors. The knowledge and management of the wild boar population and good coordination among the veterinary services, forestry authorities, and the hunting community are essential to successfully prevent, early detect, control and eventually eradicate this disease. Hunting activities in the Balkans area are regulated by national laws that set up the hunting period, the hunting grounds’ boundaries, their related management, the large and small game species to be hunted, and the tender for private offers in the case of Serbia.

There are multiple checklists and tools available to assess biosecurity, and the risk for disease introduction, detection and spread amongst farms. There is also a study that assessed the routes of ASF infection from wild to domestic pigs [17]. However, an equivalent tool to assess the risk in hunting grounds has not yet been developed. This study aims to fill this gap by developing, validating and standardizing a semi-quantitative method to assess several risks of ASF (i.e., the risks of introduction and spread into and between hunting grounds, and the risk of not detecting the infection), as well as to classify and visualize hunting areas by risk (and how such risks could be reduced if mitigations measures were applied), thus allowing those involved to efficiently prioritize corrective actions.

## 2. Materials and Methods

The flowchart of the tool, showing the steps and outputs, is presented in Figure 1. Obtaining the risk scores for each hunting ground involved several steps, described in more detail in the following sub-sections. In summary, risk factors were identified through a literature review. Out of a thorough survey filled in by hunting ground managers on biosecurity and practices at their hunting grounds, only the questions that informed the identified risk factors were selected (1 to 5 questions per risk factor). An expert knowledge elicitation (EKE) involving international experts was carried out to weigh the importance of each risk factor. Each risk factor contributes to one or more of the three assessed risks, namely the Risk of ASF Introduction, the Risk of ASF Spread and the ASF Detection Capability. These three risks are calculated by adding the relevant risk factors, each multiplied by the weight assigned through the EKE scores (i.e., the median of the scores). Finally, the Overall Risk is calculated by adding the scores of the Risk of ASF Introduction and the Risk of ASF Spread and subtracting the ASF Detection Capability score.

### 2.1. Study Area

The study targeted Kosovo, Montenegro and Serbia, all within the Balkans. Their main characteristics are summarized in Table 1.

### 2.2. Questionnaire

A questionnaire was developed to assess the biosecurity applied in hunting grounds, the behavior and practices of hunters, and management practices and characteristics of hunting grounds relevant to disease transmission (Appendix A). The questionnaire was based on a 2020 checklist developed by the Serbian competent authorities to assess the hunting biosecurity of Serbian hunting grounds that was shared with the FAO for comments, further improvement and to support online data collection. A literature review was conducted to ensure that all relevant risk factors were included.

The questions were grouped into different sections, namely: Hunting ground characteristics; Status of the hunting ground depending on the ASF risk zone; Active and passive surveillance for ASF; Domestic and wild pigs interface; Control of wild boar hunting procedures; Disinfection measures; Control of the procedures for the safe removal of offal or dead animals; Awareness; Feed ban control; and Previous controls.

In Montenegro, the questionnaire was filled in through face-to-face interviews with hunting ground managers and answers were recorded on paper and then copied into an Excel spreadsheet. In the case of Kosovo and Serbia, replies to the questionnaire were captured electronically using an online platform (Epicollect5).

### 2.3. Question Selection and Risk Factor Identification

Figure 1 represents, visually, how the questions were used to eventually provide an overall risk score. Only questions relevant for assessing the three following risks were selected from the questionnaire: the risk of ASF introduction into the hunting ground, the risk of ASF spreading to other hunting grounds, and the capability to detect ASF. The most important risk factors contributing to these three risks (37 in total) were selected, keeping in mind that not all risk factors are relevant to assess all three risks (Appendix A). As a result, each of the three risks is determined by multiple risk factors (between 12 and 24), which in turn are the result of a combination of up to five survey questions. Each selected question may contribute to one or more of the risk factors.

When assessing the Risk of ASF Introduction and the Risk of ASF Spread, the model only considered the spread into or between the hunting grounds, not into domestic pigs. Similarly, the ASF Detection Capability only refers to detection in wild boar.

At the time of the survey, Serbia had assigned risk categories to each hunting ground (i.e., infected, at risk, etc.), which requires certain additional hunting biosecurity measures to be followed by the hunting grounds. This variable was not taken into account, because even if measures are prescribed, they do not ensure that they are actually implemented in the field.

### 2.4. Expert Knowledge Elicitation Implementation

An EKE was developed to weigh the importance of the various risk factors for the three risks to be assessed (i.e., introduction, spread and detection) based on a scale from 0 to 5 (0 = not relevant; 1 = negligible; 2 = low; 3 = medium; 4 = high; 5 = critical). Experts for the EKE were selected using the snowball sampling method, i.e., a recruitment strategy in which the participants are asked to identify other potential subjects with relevant expertise [22]. Eight experts were identified as seed nominators: four experts were selected because of their field expertise (i.e., authors of guidelines, participants of international assessment missions and/or country representatives for ASF management) and four were academic experts. The level of relevant academic expertise was measured through a literature review. Search strings using relevant keywords were used on the Web of Science, Google Scholar and PubMed. The string included wild boar AND hunt* AND (disease OR African swine fever OR hepatitis E OR brucellosis OR tuberculosis OR trichinellosis OR leptospirosis OR foot and mouth disease OR classic swine fever), the latter being the most important infectious disease of wild boar. The four experts with the most publications were selected. Each expert was asked to nominate up to five other top-experts, who were contacted for another iteration of nominations. The snowball procedure consisted of four iterations (conducted between 29 January 2021 and 3 March 2021), which stopped when the number of country self-nominations, i.e., experts nominating only others within the same country, became too high (more than 80%). A network analysis was performed to understand the level of saturation reached using the package “Network” in R-Studio. The experts’ opinion was elicited through questionnaires sent via email in January and February 2020, where the experts were asked to score the risk factors (Appendix A). The questionnaire, built on Microsoft Excel (2010) spreadsheets, was organized in nine sections (same as the questionnaire structure) and pre-tested with two experts to ensure the clarity of the instructions, minimize question ambiguity, and refine the process.

### 2.5. Expert Knowledge Elicitation Statistical Analysis

Expert scores for each risk factor were reported as means, medians and standard deviation. Spearman’s rank test was used to assess the correlation between scores for perceived importance of different risk factor categories. To understand the agreement among the experts, the absolute value of the difference between scores was calculated for each pair of experts and each risk factor. The raw agreement was calculated as the percentage of comparison with total agreement (difference of 0) and percentage of deviation by different amounts between two experts (difference from 0 to 5). The Wilcoxon–Mann–Whitney test was applied to investigate differences in the median scores for the perceived importance between risk factors. The proportion of total variance was assessed as intra-class correlation coefficients (ICC) with non-parametric repeated measures ANOVA models as in Kuster et al. [23]. The outcomes were the scores on the perceived importance of each risk factor, individual biosecurity measure as subject variable, and expert as random effect. All statistical analyses were performed with R Core Team [24].

### 2.6. Risk Categorization and Mapping

The process to obtain the final risk values for each hunting ground follows several steps (Figure 1). The initial steps are described in 2.2 and 2.3. Each risk (Risk of ASF Introduction, Risk of ASF Spread, and ASF Detection Capability) was calculated by multiplying the importance score obtained through the EKE by the sum of the values that the questions bring to the risk factors. The Overall Risk was calculated by adding the Risk of ASF Introduction and the Risk of ASF Spread minus the ASF Detection Capability, where the three risks were assigned the same weight (1:1:1). To make the method replicable and comparable across countries and territories and risks, the values were normalized on a scale from 0 to 100 using the worst and best possible scores for a hunting ground. Mapping required the shape files of the hunting grounds and were colored depending on the level of risk from red (high risk) to green (low). Regarding the ASF detection capability, the color range was inverted because lower values describe a worse situation than higher values. In Kosovo, in the absence of shape files of the hunting grounds, the municipality was used instead, which generally resembled the hunting grounds. QGIS [25] was used to merge the shapefiles provided by the authorities (Montenegro and Serbia) and the NUTS 2 level shapes file from an online resource was used for Kosovo. The package tmap developed by Tennekes [26] was used to plot the data on the maps.

### 2.7. Feasibility Assessment

An additional questionnaire, Likert type items, was developed and addressed to the hunting ground managers via telephone surveys (Kosovo) or within the general survey (Montenegro and Serbia) to assess the feasibility of different mitigation measures on the hunting grounds. For each selected risk factor, a question was formulated to assess the feasibility of implementing a mitigation measure in a range from 1 to 6 (impossible = 1; very difficult = 2; difficult = 3; possible with some effort = 4; possible with little effort = 5; and possible with no effort = 6). Only risk factors with a mean score higher than 3.5 were considered for the feasibility assessment.

## 3. Results

### 3.1. Questionnaire

A total of 258 hunting grounds were surveyed. In Kosovo, a consultant and staff member of the Food and Veterinary Agency surveyed 18 hunting grounds (out of 32; 56.3%) between October 2020 and April 2021. In Montenegro, the Hunting Association of Montenegro surveyed 35 hunting grounds (out of 35; 100%) between September and October 2021. In Serbia, the Ministry of Agriculture, Forestry and Water Management surveyed 205 hunting grounds (out of 321; 63.9%) between July 2021 and January 2022.

### 3.2. Selected Questions for the Expert Knowledge Elicitation

From a questionnaire with more than 300 questions, 62 were selected because of their importance to assess the Risk of ASF Introduction, the Risk of ASF Spread, and the ASF Detection Capability. These questions contribute, in varying proportions, to a total of 37 risk factors, which were subjected to an EKE (Appendix A). Out of the 37, two risk factors, namely “*High number of visitors in the hunting ground*” and “*Absence of disinfection point for every visitor*”, were not included into the risk score, because it was not possible to collect data on the number of visitors of each hunting ground due to privacy reasons and the presence of a disinfection point is not usual in the study area.

### 3.3. Snowball Sampling Method

A total of 136 experts were referred along the snowball procedures. The minimum cut-off value (i.e., number of nominations) to include an expert in the elicitation was three. This led to the selection of 33 experts, of which 24 answered the elicitation (answer rate of 70%). For incomplete questionnaires, a reminder was sent to avoid missing values. The experts were mainly veterinarians (15), categorized as epidemiologists (5), animal health officers (2), wildlife managers (2), and not specified (6). The remaining nine experts were wildlife biologists. The experts were based in the following countries: Spain (7), Germany (4), Italy (3), Czech Republic (2), France (2), Belgium, Bulgaria, Latvia, Lithuania, Poland and Sweden (1).

### 3.4. Snowball Sampling Method

The results of the network analysis study conducted to understand the level of saturation reached is presented in Appendix A. The degree of agreement among the experts, where a score of 0 means complete agreement and 5 is a strong disagreement, is depicted in Figure 2. For Risk of ASF Introduction and Risk of ASF Spread, scores of agreements (i.e., those scored between 0 and 2) accounted for more than 90%. For the ASF Detection Capability, the scores of agreement (0 to 2) accounted for almost 80%. The scores of disagreement (4 or 5) accounted for less than 3% for the three risks.

### 3.5. Intra-Class Correlation (ICC)

The reliability (degree of correlation and agreement between measurements) was calculated for the Risk of ASF Introduction (624 measurements), Risk of ASF Spread (552) and ASF Detection Capability (312) looking into two factors: (1) The two-way random effect model chosen because of the generalization of the results to any expert who possesses the same characteristics as the selected one in each reliability study; and (2) The absolute agreement calculation used to understand the extent to which different experts assigned the same score to the same risk factor [27] (Liljequist et al., 2019). The variance percentages output from the two-way random effects model showed a smaller disagreement between the experts for Risk of ASF Introduction compared to the Risk of ASF Spread and the ASF Detection Capability. The average measure of ICC for the Risk of ASF Introduction is 0.86 (0.79–0.92), for the Risk of ASF Spread it is 0.88 (0.81–0.93) and for ASF Detection Capability it is 0.85 (0.74–0.93) (*p*-value < 0.01).

### 3.6. Expert Knowledge Elicitation

The experts’ scores for all the risk factors contributing to the Risk of ASF Introduction, Risk of ASF Spread and ASF Detection Capability are presented in Table 2.

Positive significant correlations among EKE scores were found only between the scores of Risk of ASF Introduction and that of Risk of ASF Spread. *“Shared personnel, vehicles and equipment”* showed the strongest correlation (r > 0.87), followed by “*absence of disinfection barriers”, “absence of transport or storage space used only for wild boar meat to avoid cross contamination”, “movement of wild boar”* and *“wild boar being transient”,* which were highly correlated (r > 0.8). *“Low proportion of fenced area”* had a medium correlation (r > 0.5).

### 3.7. Risk Scores and Maps

The final scores for each risk and each country or territory are depicted in Table 3 and in Figure 3B–E, showing a similar situation across the study area. Keep in mind that the higher the score, the greater the risk, except for the ASF Detection Capability, where lower scores indicate higher risk. Figure 4 highlights that the survey coverage was particularly low in Northern Serbia (Voivodina). The distribution of the Overall ASF Risk values is depicted in Figure 4, while the distribution for the three risks is provided in Appendix A. Kosovo ranked worst for three out of four risks. This is explained by the absence of training for hunting managers on ASF in wild boar at the time of the survey, the presence of shared personnel, vehicles, equipment or facilities between the hunting grounds, and the practice of supplementary feeding. Moreover, the absence of wild boar carcass submission and lack of awareness campaigns, both to the general public and hunters, about the importance of reporting dead wild boar, strongly reduced the ASF Detection Capability. In Montenegro, the higher value for the Risk of ASF Spread is explained by the sharing of personnel, vehicles, equipment or facilities between the hunting grounds. In Serbia, except for some outliers (evident in Figure 3 and Figure 4, and Appendix A), the Overall ASF Risk is the lowest of the three, while the Risk of ASF Spread ranks second because of the supplementary feeding in areas not affected by ASF, the failure to reduce wild boar populations, and the fact that pig farm workers, or hunters who have been in a pig farm in the previous 72 h (before the hunting activity), are still allowed into hunting grounds.

Additionally, there was no significant difference between the Overall ASF Risk scores of hunting grounds categorized by Serbia as “infected” and non-infected hunting grounds (i.e., described either as: “moderate risk”, “high risk” and “endangered”). This analysis was just conducted for the Serbian hunting grounds since it was the only ASF-affected country within the study area.

### 3.8. Feasibility Scores and Maps

Results from the Kruskal–Wallis test described no statistically significant difference between the feasibility scores for each question. Output of the ordered logistic regression showed that the infectious status has a highly significant effect on the scores (*p*-value < 0.05) and there is good agreement between all of the permutation tests performed (ANOVA Type, Wilks’ Lambda, Lawley Hotelling, and Bartlett Nanda Pillai Test statics) when applying a nonparametric comparison of multivariate samples as reported in Burchett et al. [28].

The probability that a random sample from the feasibility answers for specific implementation measures, namely “no kitchen waste for feeding, a plan for dead wild boar disposal, decrease wild boar density, not allowing meat inside the hunting ground and forbid driven hunt”, would give the same answer is higher than 80% when comparing the outcomes by the infectious status at country level, i.e., Serbia infected. The measures about cleaning and disinfection procedures and shared personnel and vehicles scored a low probability (<50%).

Regarding specific outcomes by country or territory, in Kosovo, movement of wild boar and sharing personnel were not included in the analysis because it was already enforced by law and the answers were all 6s (i.e., possible with no effort). Additionally, adopting leak-proof bags to transport carcasses or offal was considered to be very difficult by 81.2% of the respondents. Having a disposal plan in case of an ASF outbreak was considered possible with little effort by 68.8%. Cleaning and disinfecting vehicles before they leave the hunting ground and avoiding kitchen waste or swill for feeding was considered difficult by 65% of the hunting ground managers.

In Serbia, decreasing wild boar density through increased hunting pressure and avoiding supplementary feeding was considered possible with little effort by 60% of the hunting ground managers. Personal disinfection after dressing wild boar and avoiding sharing vehicles and personnel between the hunting grounds was considered difficult by half. Conversely, avoiding kitchen waste and other sources of animal protein to feed wild boar was considered possible with little effort by 70%.

Hunters in Montenegro appeared less willing to cooperate in terms of implementing sanitary measures and changing their general behavior. Forbid driven hunts, not allowing visitors to bring meat into the hunting ground, and forbid pig workers or pig owners to hunt were considered measures impossible to implement by more than 80% of the hunters. Decreasing the density of the wild boar population and avoiding the use of kitchen waste or other protein to feed the wild boar was considered very difficult by the 83% of the hunters.

Table 4 presents the answers (median values) of the hunting ground managers to the feasibility survey. The distribution of the scores given by the hunting managers are provided in Appendix A. Additional normalized (scale 1 to 100) maps (Figure 3F and Appendix A) were created to visually represent the changes in the risk if interventions were carried out in view of the feasibility indicated by the hunting ground managers (Table 4). The results of the feasibility study allowed us to establish three categories of interventions based on the effort required, i.e., high, medium or low. Furthermore, having three scenarios allows one to visualize the potential result to be achieved, accounting for the effort implemented, which will depend on the political will, as well as the economic and logistical capabilities. Results show that, with low effort, it is possible to dramatically decrease the ASF risk in the study region (Figure 3F and Appendix A).

## 4. Discussion

### 4.1. Results

The control and eradication of ASF in wild boar is and will likely remain a key component of the overall ASF control strategy in Europe, as well as in other parts of the world where wild boar or feral pigs are present, whether naturally or introduced. Until there is a commercially available vaccine for wild boar, close coordination and collaboration with the hunting sector and wildlife authorities remains the main strategy to limit the spread of the disease. A better understanding of how hunting grounds are managed and their biosecurity level will allow us to identify the strengths and gaps of the ASF management in wild boar populations and to tailor the mitigation measures that could reduce the risk. This is the purpose of the tool presented in this paper.

The tool was used to assess the hunting grounds in Kosovo, Serbia and Montenegro. Our study found that, currently, the hunting grounds within the study region were at high risk of ASF, specifically the risks of ASF introduction and spread, and the risk that ASF will not be detected. Kosovo had the highest risks overall, with Serbia scoring best. The latter can be partially attributed to the fact that Serbia is already an ASF-infected country and, therefore, a series of mitigation measures have already been implemented. Surprisingly, no significant difference was found between the risk scores in ASF-infected and ASF-free hunting grounds in Serbia, perhaps showing that measures were implemented across the whole country.

The assessment allowed us to identify the most common gaps across the study region, but also specific ones. In Serbia, both active and passive surveillance for ASF in wild boar are in place. The number of found dead wild boar has increased when comparing the results of 2021 and 2022; however, the ASF-affected hunting grounds contribute most of the found wild boar and samples submitted, while unaffected areas have a low submission rate. Montenegro also relies on passive surveillance, but the overall number of found dead wild boar per year is very low compared to the wild boar population of the country. In Kosovo, at the time of the study, there was no systematic effort to detect the disease in wild boar.

On the spot evisceration of hunted wild boar is still widely practiced. In the case of Serbia, more than half of the hunting grounds surveyed have a temporary game meat storage to store meat until the PCR test results arrive. In Montenegro and Kosovo, none of the hunting grounds have game meat storage facilities. Only in Serbia is disinfection widely used to clean and disinfect facilities. The majority of hunters regularly disinfect their hands, clothing, footwear and equipment after the manipulation or handling of offal or dead animals (72% in Kosovo, 83% in Montenegro and 89% in Serbia).

Supplementary feeding of wild boar is still a general practice in ASF-free hunting grounds. Additionally, there is no increased hunting pressure nor selective hunting of adult or subadult females to decrease the wild boar population.

Based on the key findings and the feasibility study, a thorough written report with recommendations and maps was provided to each country.

### 4.2. Uses of the Tool

The tool allows users to visualize the different ASF risks of hunting grounds, whether as numerical values or maps, at sub-national, national and regional levels. The different maps (i.e., risks and feasibility) can be used to guide policy makers by highlighting which aspects of hunting biosecurity and management, or which geographical areas should be prioritized:identify the weakest hunting grounds (i.e., those at higher risk).categorize the hunting grounds based on their Overall ASF Risk score.pinpoint where the ASF surveillance in wild boar is weakest, i.e., where the uncertainty of freedom of disease is the highest.identify the easiest or cheapest (i.e., most feasible) mitigation measures to decrease the ASF risk, whether at individual hunting grounds or the whole country.compare the level of hunting biosecurity between countries or regions.monitor the change of risk over time, by repeating the surveys and analyses.guide decision making at hunting grounds during outbreaks, by prioritizing actions not just because of proximity, but also at those hunting grounds with high-risk scores.

### 4.3. Limitation and Biases

Implementing the surveys is labor-intensive. When there are many hunting grounds, data collection can take a long time. However, data collection should be completed in the shortest possible time, especially if ASF is already present in the country or region. When a hunting ground is affected with ASF, the relevant competent authorities will likely implement measures, which will change the risk scores.

The tool was designed to be implemented in certain hunting management systems, more specifically, in Eastern Bloc countries. Therefore, if planning to use the tool in other countries with different systems, e.g., in Western Europe, certain modifications and adaptations will be necessary. Furthermore, such adaptations might make it impossible to compare the results with other countries.

The risk factors identified and scored during the EKE are mostly generic and could be extrapolated to other geographical regions. However, they were selected with European settings in mind and had a high degree of accordance with the risk factors found by de la Torre et al. [29]. Some risk factors specific to other geographical regions may not have been considered in our tool. Moreover, all the experts contributing to the EKE were based in Europe.

This tool does not directly consider the ASF status (whether known, unknown or consciously hidden). However, it is already reflected in the type and stringency of the measures implemented. This could be one of the reasons why Serbia (the only infected country) scored better, i.e., because they are taking more and stricter measures.

Finally, while the questions in the hunting ground survey are objective, the feasibility study is subject to the personal views of the hunting ground managers interviewed. While there could be a conflict of interest when filling in the survey, the fact that the results are in line with expectations (i.e., high risks throughout) suggests that the hunting ground managers were truthful when answering and critical of the gaps and limitations of their hunting grounds.

### 4.4. Future Work

Future work on the tool will aim to assess additional countries. A guideline will be developed to provide instructions for each step of the workflow together with various templates for key steps. Work to improve the tool will continue to enhance the user experience. First, to incorporate internal logical checks within the electronic questionnaire to verify the consistency of answers given and to highlight errors. Second, to automate calculations so that risk scores become immediately available as soon as the surveys are filled in. The third and most ambitious option is to develop a web-based interface so that data providers can select their preferred language, have immediate access to the questionnaire and obtain their risk scores immediately. If hunting ground shape files are available, the scores could also be immediately visualized.

## Figures and Tables

**Figure 1 pathogens-11-01466-f001:**
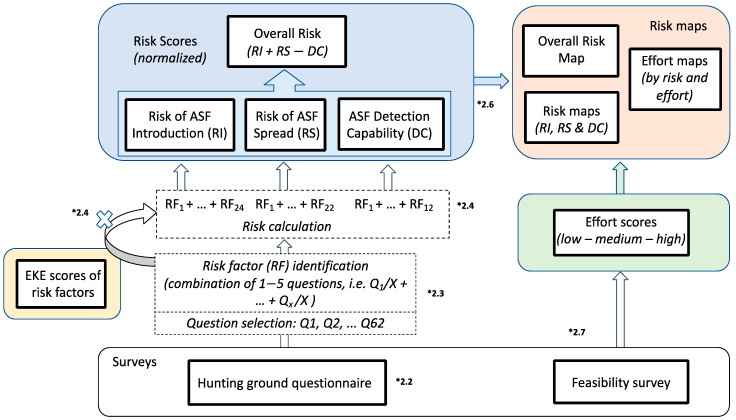
Flowchart of the tool used to assess the risk for ASF in hunting environments. Numbering (*) refers to the section of the paper that describes the step. Abbreviations: Q—question; EKE—expert knowledge elicitation; RF—risk factor; RI—Risk of ASF Introduction; RS—Risk of ASF Spread; DC—ASF Detection Capability.

**Figure 2 pathogens-11-01466-f002:**
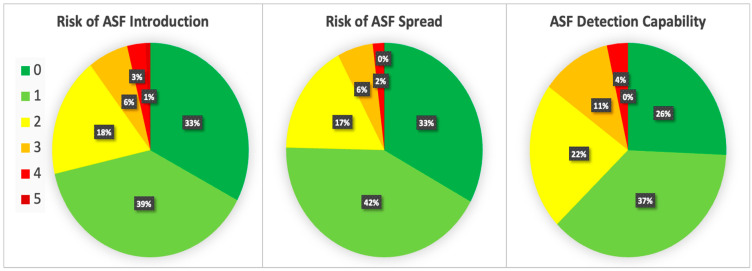
Percentages of agreement between experts for each risk factor.

**Figure 3 pathogens-11-01466-f003:**
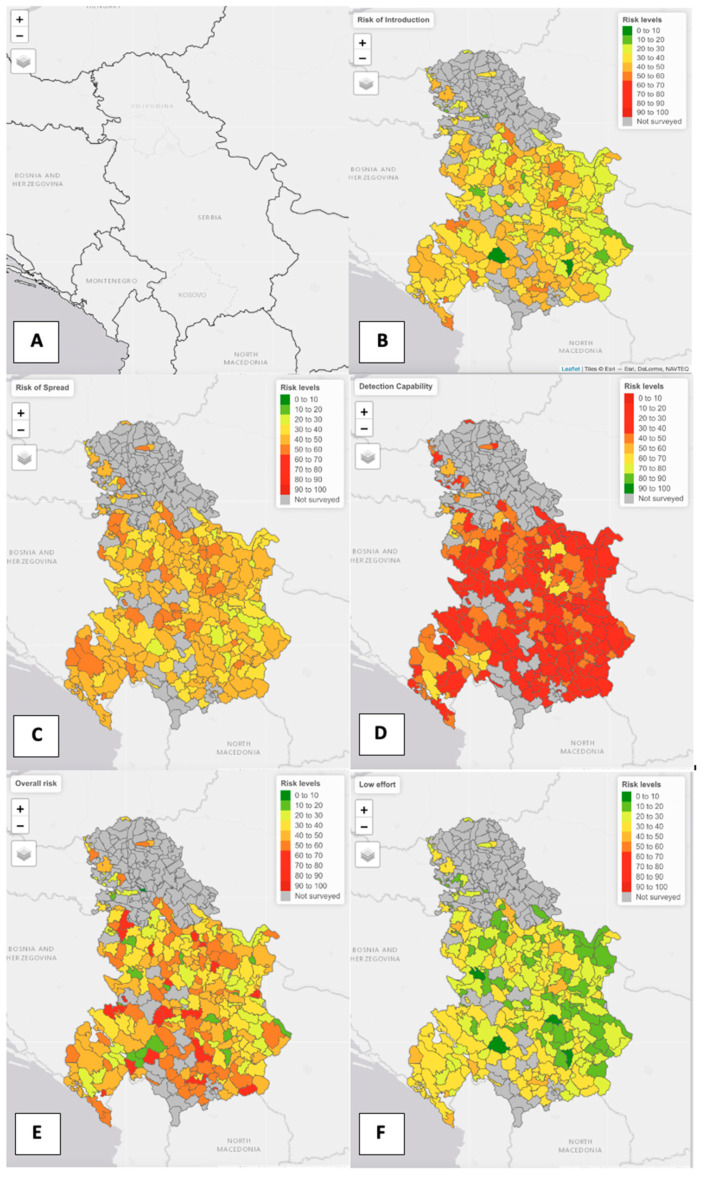
Maps showing (**A**) the study region (Kosovo, Montenegro and Serbia); (**B**) the Risk of ASF Introduction; (**C**) the Risk for ASF Spread; (**D**) ASF Detection Capability; (**E**) the Overall Risk of ASF, and; (**F**) the Overall Risk of ASF if low effort mitigation measures are applied.

**Figure 4 pathogens-11-01466-f004:**
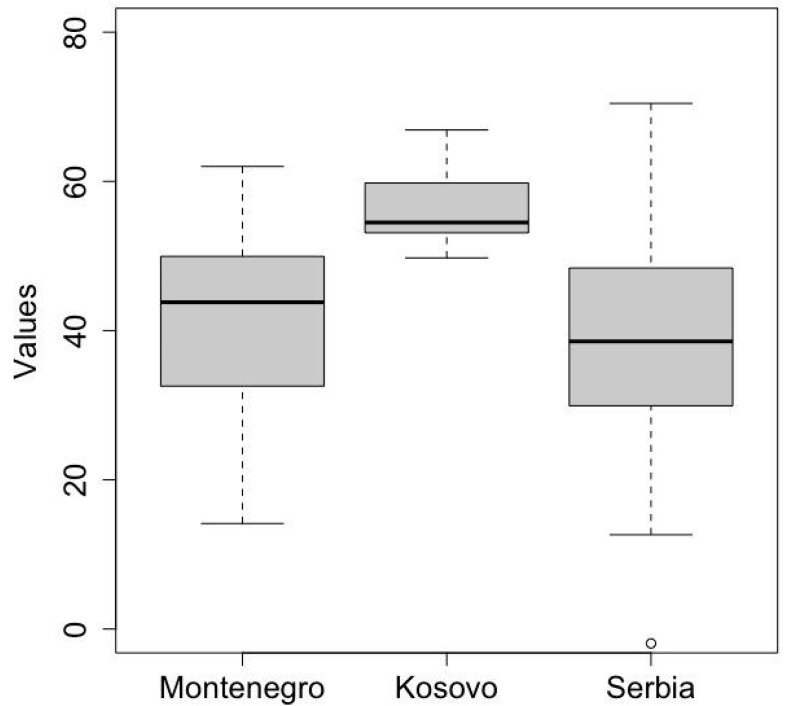
Distribution of Overall Risk scores.

**Table 1 pathogens-11-01466-t001:** Main characteristics of the countries and territories under the study.

Country/ Territory	Human Population	Country/ Territory Extension	Suitable Area for WB	Number of HG	Hunting Season	ASF Status	WB Population	WB Density
Kosovo ^1^	1.8 mln	10,908 km^2^	82.5% [9]	32	1 July–31 January	Free	<2000 [18] 3000–3500 [19]	1.34 per km^2^ [18]
Montenegro	0.6 mln	12,812 km^2^	82.0% [9]	35	1 October–31 January	Free	5616 [20]	0.20–0.33 per km^2^ [9]
Serbia	7.0 mln	88,407 km^2^	75.3% [9]	321	1 April–31 March	Infected	25,606 [21]	0.47–0.75 per km^2^ [9]

Abbreviations: WB—wild boar; HG—hunting ground; mln—million.

**Table 2 pathogens-11-01466-t002:** Mean, median (MDN) and standard deviation (SD) values results from the expert knowledge elicitation (EKE) for ASF risk factors.

Risk Factor	Risk of ASF Introduction			Risk of ASF Spread			ASF Detection Capability		
	Mean	Mdn	SD	Mean	Mdn	SD	Mean	Mdn	SD
Supplementary feeding of WB with products of animal origin from uncontrolled sources (swill or kitchen waste)	4.75	5	0.53						
Movements of live WB between HGs. This refers only to WB moved or transported by humans	4.38	5	0.92	4.38	5	0.97			
Access to landfills or scavenging places for WB	4.25	4	0.79						
High number of pigs in low biosecurity farms (e.g.,: free range or backyard) present in HG	4.21	4.5	1.18	3.33	3	1.01	3.04	3	1.27
Lack of compliance by the HG user, i.e., the user has not applied measures ordered by the authorities in the past 2 years	4.08	4	0.78	4.08	4	0.78	2.71	3	1.63
Bringing meat products into the HPA is allowed	4.08	4	0.97						
High WB density in the HPA	3.92	4	1.02	4.50	5	0.78	3.92	4	1.18
Shared personnel vehicles, equipment and facilities (e.g., dressing and storage areas) with other hunting ground/s	3.92	4	1.02	3.92	4	1.02			
Hunters who own pigs or work on pig farms	3.88	4	1.12						
People who visited other hunting grounds or pig farmsin the last 72 h are allowed to hunt	3.71	4	1.12						
Absence of cleaning and disinfection of hands, footwear, clothing and personal equipment after offal and manipulation of dead animals	3.54	4	1.47	4	4	0.88			
Driven hunt is practiced	3.54	4	1.1	4.04	4	0.86	3.04	3	1.12
Absence of posters/flyers/briefings to inform the public and hunters upon entering the HPA about the importance of ASF and measures to prevent it	3.5	4	1.02	3.54	3	0.88	3.58	4	1.32
High number of visitors into the HG	3.5	3.5	1.10						
HPA is in contact with cities and settlements	3.46	3	0.88	2.92	3	0.97	3.42	3	1.14
Absence of disinfectants and cleaning and disinfectionprocedures for equipment, facilities, meat storage, evisceration sites	3.38	4	1.35	4.04	4	0.91			
Supplementary feeding of WB	3.33	3.5	1.27						
HPA is in contact (i.e., crossed or bordered) with roadsand/or navigable watercourses	3.25	3	0.79	3.21	3	0.83	2.88	3	1.30
Absence of disinfection point for every visitor	3.13	3	1.08						
Low proportion of fenced area within the HPA	3.08	3	1.21	3.29	3	1.08			
Vehicles used for hunting that leave the HG withoutbeing cleaned and disinfected first	3	3	1.44	3.83	4	1.13			
Large size of the HPA	2.96	3	1.3						
Absence of disinfection barriers	2.92	3	0.97	3.17	3	0.96			
Absence of transport and/or storage space used only for WB meat in order to avoid cross contamination of meat from other species	2.54	2	1.59	2.58	3	1.5			
WB being transient (occasional or seasonal game)	2.50	2	1.18	2.83	3	1.31			
HG not collecting information on the last hunting activity of hunters	2.17	2	1.55						
Training of the HG manager in game pathology and handling of game and game meat				2.88	3	1.23	3.50	3.5	1.02
Absence of fixed (designated) dressing areas				3.50	4	1.14	2.08	2	1.56
Absence of a disposal plan in case of an ASF outbreak				4.54	5	0.83			
Limited disposal of found dead WB				4.33	5	0.87			
Absence of cargo transport or leak-proof bags or tanks to move dead WB				4.21	4	0.88			
Meat or trophies leaving the hunting ground without an ASF PCR negative test				4.04	4	0.86			
No systematic destruction of offal (from hunted animals)				4.04	4	0.91			
Active search for dead WB carcasses							4.71	5	0.55
Sampling of hunted WB for ASF testing							3.63	4	1.28
Low effort or incentive in finding dead WB carcasses in the HG							2.83	3	1.49
Absence of culling							2.67	2.5	1.05

Gray cells represent risk factors not scored because they did not contribute to those specific risks. Abbreviations: Mdn—median; WB—wild boar; HG—hunting ground; HPA—hunting productive area (defined as the part of the hunting ground where the hunting game species has favorable conditions for life, reproduction, proper development, and constant survival).

**Table 3 pathogens-11-01466-t003:** Final risk scores for Kosovo, Montenegro and Serbia and confidence intervals.

	Kosovo	Montenegro	Serbia
Risk of ASF Introduction	44.28 (42.25–46.31)	40.37 (38.41–42.33)	33.62 (32.28–24.95)
Risk of ASF Spread	40.90 (39.05–42.74)	45.47 (43.51–47.42)	41.85 (40.85–42.84)
ASF Detection Capability	29.11 (27.45–30.76)	44.54 (40.88–48.20)	36.47 (35.12–37.81)
Overall ASF Risk	56.07 (53.60–58.53)	41.29 (36.72–45.86)	39.00 (37.25–40.75)

Note: The higher the score, the greater the risk, except for the ASF Detection Capability, where lower scores indicate higher risk.

**Table 4 pathogens-11-01466-t004:** Feasibility scores (i.e., medians) per country or territory.

Feasibility Variable	Kosovo	Montenegro	Serbia
How feasible is it to dispose of all or most found dead WB?	3	1	5
How feasible is it to forbid visitors and hunters from bringing meat products into the HG’s HPA?	2	1	4
How feasible is it to forbid from hunting those hunters who own pigs or work on pig farms?	4	1	3
How feasible is it to forbid driven hunts?	4	1	4
How feasible is it to forbid hunters who have visited pig farms in the last 72 h from hunting?	4	1	4
How feasible is it that the HG management does NOT use kitchen or restaurant waste or other sources of animal protein (swill) to feed WB?	3	2	4
How feasible is it to create a plan for how and where to dispose of dead WB in case of an ASF outbreak?	4	2	4
How feasible is it to decrease the density of WB by increasing hunting, eliminating supplementary feeding or other means?	4	2	4
How feasible is it to prevent WB accessing landfills or scavenging places?	2	2	4
How feasible is it NOT to share any personnel, vehicles, equipment or facilities (such as dressing or storage area) with other hunting grounds?	6	2	4
How feasible is it NOT to transport live WB into or outside of your HG?	6	3	4
How feasible is it to have leak-proof bags or tanks available on stock to transport dead WB carcasses out from the HG?	2	3	4
How feasible is it to inform the public and hunters (through posters, flyers or face-to-face briefings) about the importance of reporting dead WB and about the main ASF prevention measures?	4	3	4
How feasible is it to destroy all offal from hunted WB?	4	3	4
How feasible is it to establish fixed areas to dress carcasses?	4	3	4
How feasible is it to always clean and disinfect the hands, footwear, clothing and equipment after dressing and/or handling dead animals?	3	4	4
How feasible is it to always clean and disinfect vehicles before they leave the HG?	3	4	4
How feasible is it to always clean and disinfect hunting equipment, facilities, meat storage or dressing areas after they are used?	3	4	3
Average of the medians	3.61	2.33	3.94

Abbreviations: WB—wild boar; HG—hunting ground; HPA—hunting productive area (defined as the part of the hunting ground where the hunting game species has favorable conditions for life, reproduction, proper development, and constant survival). Scale used: impossible = 1; very difficult = 2; difficult = 3; possible with some effort = 4; possible with little effort = 5; and possible with no effort = 6.

## Data Availability

Data supporting the results can be found in the Appendix A.

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
