# Peer review of "A Novel Tool to Assess the Risk for African Swine Fever in Hunting Environments: The Balkan Experience"

_pathogens, 2022, doi:10.3390/pathogens11121466_

Round 1

Reviewer 1 Report

The work addresses an important topical issue and focuses attention on a context in which the problem is present and may even worsen in the near future. It is in fact ASF in the Balkan region: the disease has already been reported, is probably severely underestimated and shows a tendency to become endemic and spread to neighbouring territories or to more distant ones through the human factor.
The aim of the study is clear: it is intended to propose a new tool for risk analysis and also to verify the effect that the adoption of measures aimed at mitigating some of the most impactful risk factors would have.
The tools used for the work are not particularly original; if anything, the approach used to recruit the experts on which to base the analysis is slightly different. All this makes the work not very interesting and innovative.
Furthermore, in my opinion there is an important shortcoming: all the work is largely based on international experts, who inevitably have little knowledge of the Balkan reality, or on local stakeholders who are probably influenced by various situations (conflicts of interest, poor training, low awareness...). The entire survey could be influenced by these basic shortcomings and in fact the results appear quite predictable and in line with expectations.
In general terms, the working method is well described and rather interesting supplementary materials are also provided, but it would be appropriate if at the beginning of the chapter Materials and Methods the authors explained the logical procedure used more clearly. In the current version, everything is delegated to a flow chart which, although clear, is not sufficient; it is true that the method is subsequently recalled and explained, but an organic revision of these explanations is needed.
In the discussion chapter,  it would be appropriate for the authors to address the problem of how the chosen respondents might not be completely clear about the situation.

In my opinion there is also an important problem: if we consider that the available data lead to an underestimation of the incidence and prevalence of ASF in the Balkans, the authors ignore the issue. The available information that verosimilarly is deficient and therefore inappropriate may have in fact conditioned the opinions of experts and stakeholders undermining the reliability of all the work done. this critical issue should be addressed altirmenti the work is exhausted with a mere and onerous data collection and steile statistical analysis. The possibility that the method is appropriate, but the information collected is not atrtendible enough should be properly presented to decision makers given that the goal of the work is to provide decision-making elements.

Author Response

The authors would like to thank the reviewer for the excellent comments. Are responses and clarifications can be found in the attached (in red).

Reviewer 2 Report

In Europe, African swine fever (ASF) can be maintained in wild boar populations, which is a constant source of the virus and a major challenge in the management of this disease. In this study, risk factors were weighted according to expert opinion, and a new survey-based tool was developed to assess the risk of ASF in hunting areas and to show how the risk would change if the measures were applied. This is a fairly large scale study over a long period of time. Generally speaking, it is not an easy field to gain knowledge in, as the support from not only the authors but also their collaborators is indispensable. Therefore, I think the significance of this study is great.

I have some minor comments. Please see below.

Generally, for risk assessments, indicate in the paper whether the methodology is quantitative, qualitative, or semi-quantitative. Please consider clearly stating this somewhere in your manuscript.

L39: Delete the double space.

L64-67: Since this seems to be the opinion of the authors, please consider moving sentences from Introduction to Discussion. If you want to keep it in the introduction, I suggest you change it to something objective.

L198-199:

If I am not wrong, as far as I can see from Supplementary File 3, some of the questionnaire items in calculating the three risks (RI, RS, DC) are overlapped. I think that is fine when calculating each risk (RI, RS, DC independently), but do you consider the double counting of the risk when calculating the overall risk?

When you think about sets A and B, there is an intersection (AB). In this case, the overall risk is obtained as follows “Overall = A + B - AB”

Were the three risks considered 1:1:1 in importance when determining the final risk score? If so, it would be better to state a word about it. This is because depending on how the final risk is defined, these three risks can be weighted (ex. RI:RS:DC = 2:3:1).

In section 3.7: There seems to be a mix of results and considerations. I suggest you reread it again and consider moving some of it to the discussion part.

L312: Missing “)”

Figure 4: Please describe how you created figure 4. did you use GIS software? Also, please mention the source of the shp layers for the study area in section 2.6.

Author Response

The authors would like to thank the reviewer for the excellent and constructive comments. Are responses and clarifications can be found in the attached (in red).
